# Assessing the Levels of Robusta and Arabica in Roasted Ground Coffee Using NIR Hyperspectral Imaging and FTIR Spectroscopy

**DOI:** 10.3390/foods11193122

**Published:** 2022-10-07

**Authors:** Woranitta Sahachairungrueng, Chanyanuch Meechan, Nutchaya Veerachat, Anthony Keith Thompson, Sontisuk Teerachaichayut

**Affiliations:** 1Department of Food Science, School of Food-Industry, King Mongkut’s Institute of Technology Ladkrabang, Chalongkrung Road, Bangkok 10520, Thailand; 2Department of Food Process Engineering, School of Food-Industry, King Mongkut’s Institute of Technology Ladkrabang, Chalongkrung Road, Bangkok 10520, Thailand; 3Department of Postharvest Technology, Cranfield University, College Road, Bedford MK43 0AL, UK

**Keywords:** qualitative, quantitative, classification, detection, spectra

## Abstract

It has been reported that some brands of roasted ground coffee, whose ingredients are labeled as 100% Arabica coffee, may also contain the cheaper Robusta coffee. Thus, the objective of this research was to test whether near-infrared spectroscopy hyperspectral imaging (NIR-HSI) or Fourier transform infrared spectroscopy (FTIRs) could be used to test whether samples of coffee were pure Arabica or whether they contained Robusta, and if so, what were the levels of Robusta they contained. Qualitative models of both the NIR-HSI and FTIRs techniques were established with support vector machine classification (SVMC). Results showed that the highest levels of accuracy in the prediction set were 98.04 and 97.06%, respectively. Quantitative models of both techniques for predicting the concentration of Robusta in the samples of Arabica with Robusta were established using support vector machine regression (SVMR), which gave the highest levels of accuracy in the prediction set with a coefficient of determination for prediction (R_p_^2^) of 0.964 and 0.956 and root mean square error of prediction (RMSEP) of 5.47 and 6.07%, respectively. It was therefore concluded that the results showed that both techniques (NIR-HSI and FTIRs) have the potential for use in the inspection of roasted ground coffee to classify and determine the respective levels of Arabica and Robusta within the mixture.

## 1. Introduction

Several factors can influence the flavor of brewed coffee, including: the climate, environment and soil where the plants are grown, as well as the harvesting method and maturity of the beans, but a major influence is the species from which the raw beans are obtained [1,2]. There are more than 70 species of *Coffea*, with Arabica (*C. arabica*), which originated in the Ethiopian highland, and Robusta (*C. canephora*), which originated in central and western sub-Saharan Africa, being the main cultivated species. Arabica accounts for approximately 60 to 70% of the total cultivated worldwide. The coffee made from these two species has different characteristics and beans can be identified visually, but it is not possible to differentiate between them visually when they have been ground. The price of Arabica beans is approximately twice that of Robusta beans; therefore, branded ground coffee or instant coffee can be a target of fraud, often through the partial or complete substitution of Arabica with Robusta [3]. Reference [4] tested 22 roasted ground coffee samples, labeled 100% Arabica coffee, from 12 different brands from shopping malls and coffee stores in Taiwan and found that four samples were adulterated with Robusta coffee. Moreover, ground coffee has been the target of fraudulent admixtures with cheaper materials, including coffee husks and other roasted grains [5] and roasted barley [6]. Many commercial brands of ground coffee contain blends of Arabica and Robusta, which are blended in order to achieve a flavor typical of that particular brand at a price that optimizes the market for that brand. Arabica has half the caffeine of Robusta, but it is a delicate plant that requires more attention and care and is sensitive to pests and diseases. It is also cultivated in higher altitude land (in Colombia coffee for export can only be grown by law between 1200 and 1800 m above sea level). Robusta is grown at heights between 200 and 800 m above sea level, does not require as much care as Arabica, is more resistant to pests and diseases and has a higher yield per plant.

The NIR hyperspectral imaging (NIR-HSI) technique combines spectroscopy and computer vision techniques in order to provide both spectral and spatial data that associates with the characterization, composition and properties of the samples [7]. The hyperspectral imaging technique can be applied as a fast, non-destructive method to detect contamination of powdered food [7,8]. Previously, research has reported on using NIR-HSI for evaluating the adulteration of powdered food such as prediction of black pepper adulteration [9,10], adulterations in wheat products with cheap grains [11], identification and quantification of adulterants in milk powder [12,13,14], detection of adulteration of peanut flour in chocolate powder [8] and prediction of concentration of adulterants in tapioca starch [15].

Fourier transform infrared (FTIR) spectroscopy is one type of infrared spectrometry that uses the principle of absorption energy of molecules based on their vibrations. Different molecules have different spectra [16]. This method acquires the spectral information of samples in the infrared wavelength range and measurement is fast, non-destructive, accurate, reliable [16] and does not use any chemicals [17]. FTIR spectroscopy can be used for both qualitative and quantitative analyses [18] and has been used both to identify chemical components in samples and to detect contaminants in samples [19]. In addition, it can be used for the analysis of adulteration in food [17]. FTIR spectroscopy was successfully used for detecting adulteration in foods such as prediction of adulteration of starch in onion powder [18], adulteration both qualitative and quantitative in tea [20,21], adulteration in paprika powder [22] and adulteration in garlic powder [23,24]. In summary, both NIR-HSI and FTIR spectroscopy techniques have been successfully used for detecting adulteration in foods. Therefore, the aim of this research is to compare the performance of both techniques for evaluating the adulteration in Arabica ground coffee with Robusta ground coffee based on classification and calibration models. This will be useful for the industries in order to consider a suitable technique to inspect the raw material of the ground roasted coffee before feeding it to the processor.

## 2. Materials and Methods

### 2.1. Sample Preparation

The Arabica roasted coffee beans were purchased from a coffee plantation in Chiang Rai province in northern Thailand and the Robusta roasted coffee beans from a coffee plantation in Chumphon province in southern Thailand and then ground in a blender (Blender 480, Kenwood, Thailand). Samples of Arabica were mixed with samples of Robusta at a ratio of 1% to 99% (*w*/*w*) and increasing every 1% (*w*/*w*) until the ratio was 99% to 1% (*w*/*w*). Each of the samples was stored separately in a zip-lock plastic bag at 25 °C until required for NIR-HSI and FTIRs measurements.

### 2.2. NIR-HSI and FTIRs Data Acquisition

Each sample in a container was scanned using NIR hyperspectral imaging (Specim FX17e, Spectral Imaging Ltd., Oulu, Finland) in an air conditioned room at 25 °C (Figure 1A). Scanning was performed by 224 spectral bands in the wavelength range of 935–1720 nm with an interval of 3.5 nm in the reflectance mode with a scanning speed of 15 mm/s. Two references were used: a black reference (R_b_) by closing the shutter and covering it with a lid on the camera lens, and a white reference (R_w_) using a rectangular Spectralon bar. 

Each sample was placed in an attenuated total reflectance (ATR) crystal and then scanned using an FTIR spectrophotometer (Bruker Corporation, INVENIO-S, Ettlingen, Germany) with a DLaTGS detector in the air conditioning room at 25 °C (Figure 1B). The absorbance spectra were collected using 64 scans in the wavenumbers of 4000 cm^−1^ to 400 cm^−1^ with the interval of 4 cm^−1^ in the reflection mode and the average spectrum of each sample, contained with 2519 independent variables, was acquired. 

### 2.3. Principal Component Analysis (PCA)

PCA was used for verifying the possibility of discrimination between two groups based on spectral information of both NIR-HSI and FTIRs. Additionally, PCA was carried out for comparison between samples of pure Arabica and pure Robusta as well as between samples of pure Arabica and Arabica with Robusta.

### 2.4. Support Vector Machine (SVM)

Samples from each of the measurements were divided between two sets. One set was used for establishing the models, while the other was used for testing the models. For qualitative analysis, the dependent variables were 0 (pure Arabica) and 1 (Arabica diluted with Robusta), and the independent variables were spectral data. For quantitative analysis, the dependent variables were the concentration levels of Robusta in samples, and the independent variables were spectral data. Before creating the models, the acquired original spectra of samples in the calibration set were preprocessed using spectral pretreatment methods (smoothing, first derivative, second derivative, MSC, SNV) and combined methods in order to obtain the optimum models for predicting both qualitative and quantitative analyses.

Previously, methods have been successfully used to establish the model from spectral data, including partial least squares regression (PLSR) [18,25], multiple linear regression (MLR) [26], principal component regression (PCR) [27] and support vector machine (SVM) [28]. Normally, PLSR is used for analysis since it has been shown to give better results when compared to MLR and PCR [29,30]. However, some research reports indicate that when SVM and PLSR were compared for qualitative and quantitative analysis [31] the adulteration of melamine in liquid milk was detected and they found the SVM model showed better results than the PLSR model. Ref. [32] used the SVM model for determining authentication of Spanish protected designation of origin (PDO) wine vinegars that the SVM model provided a better result in discrimination than those of the PLS-DA model. Ref. [33] tried to use the SVM and PLSR model for predicting the physicochemical properties of soybean paste. The results showed the SVM models gave more precisely than those of the PLSR models. Thus, SVM was selected for establishing the models for comparison of the two techniques in this study.

SVM is a machine learning technique that is used for classification and regression analysis depending on statistical principles [34]. SVM models are developed based on calculating a distance metric among data vectors by the concept of the structural risk minimization induction principle therefore it can obtain a good solution in high dimensional spaces that helps to increase the efficiency of the model [35,36]. It can be applied to establish models for both SVMC and SVMR [34].

For qualitative analysis, SVMC is a supervised method of classification that uses separate categories for two classes [37]. Qualitative models were created for classifying the dependent variable that the number of 0 was a representative of a class of pure Arabica while the number of 1 was a representative of a class of adulterated Arabica with Robusta. The independent variables were spectral data at various wavelengths and wave numbers for NIR-HSI and FTIRs, respectively. SVMC was used for developing the classification models for discriminating between pure Arabica and adulterated Arabica with Robusta in this study. Spectral data in the calibration set were preprocessed using various spectral pretreatment methods. The best models were selected by cross-validation. To evaluate the predictive ability of the models, the models were tested by the samples in the prediction set. The acquired predicted values from the models were determined by comparing them with the actual values. The result of each sample was presented as positive true, negative true, positive false or negative false. The classification models were evaluated for accuracy by Equation (1), specificity by Equation (2), sensitivity by Equation (3) and error rate by Equation (4). Accuracy is used to present the overall accuracy of the classification model. Both specificity and sensitivity are used together to present the predictive performance of the classification model. Finally, the error rate is used to present to misclassify of overall prediction of the classification model [38,39]. The same procedures in
(1)$$\mathrm{Accuracy}\;(\%)=\frac{\left(TP+TN\right)}{Total}\times 100$$
(2)$$\mathrm{Specificity}\;(\%)=\frac{TN}{\left(TN+FP\right)}\times 100$$
(3)$$\mathrm{Sensitivity}\;(\%)=\frac{TP}{\left(TP+FN\right)}\times 100$$
(4)$$\mathrm{Error}\;\mathrm{rate}\;(\%)=\frac{\left(FP+FN\right)}{Total}\times 100$$
where: *TP* = true positive, *TN* = true negative, *FP* = false positive, and *FN* = false negative.

For quantitative analysis, SVMR is a regression analysis of SVM which is used to establish the calibration model that correlated between the spectra data and the dependent variables [40]. Quantitative models were created for predicting the dependent variable, which was the concentration of Robusta in the samples. The independent variables were spectral data at various wavelengths and wave numbers for NIR-HSI and FTIRs, respectively. SVMR was used for developing the models for predicting the concentration of Robusta in this study. Spectral data in the calibration set were preprocessed by various spectral pretreatment methods. The best models were selected by considering the results of cross-validation using the coefficient of determination (R_cv_^2^) and root mean square error (RMSECV). To evaluate the predictive ability of the models, the models were tested by the samples in the prediction set. The accuracies of the models were considered by the coefficient of determination for prediction (R_p_^2^) and root mean square error for prediction (RMSEP).

The Prediktera Evince software version 2.7.9 and the OPUS program version 8.5.29 was used for device control and data acquisition for NIR-HSI and FTIRs, respectively. Statistical results were analyzed by the Unscrambler X 10.4.

## 3. Results and Discussion

### 3.1. Average Spectra Obtained from the NIR-HSI and FTIRs

The spectral image (640 × 1185 pixels) in the wavelength range of 935–1720 nm was obtained from scanning with the NIR-HSI technique. Each spectral image of the samples was removed from the spectra of the background and the edge of the container, and then the spectra of each sample were averaged. The average spectrum of each sample in the full wavelength range was used for both qualitative and quantitative analysis. Additionally, the acquired spectra of each sample from FTIRs in the wavenumbers of 4000 cm^−1^ to 400 cm^−1^ were averaged and then the average spectrum of each sample in the full wave numbers was also used for both qualitative and quantitative analysis.

### 3.2. Characterization of Spectra for Ground Roasted Coffee Using NIR-HSI

The average original absorbance spectra and 2nd derivative absorbance spectra of pure Arabica and pure Robusta by NIR-HSI in the wavelength of 935–1720 nm (Figure 2), Show only a few peaks in the original absorbance spectra. The absorbance peaks were observed at around 1200 and 1450 nm, which are related to the overtone vibrations of the second overtones and the first overtones of OH stretching, respectively, as shown in Figure 2A and previously reported by [41,42]. The original absorbance spectra were preprocessed with the 2nd derivative technique, which reduced overlaps of peaks in the spectra. The 2nd derivative spectra were used to identify the main components of both Arabica and Robusta. The feature of both spectra of pure Arabica and pure Robusta showed peaks at the same wavelength. Those peaks of the 2nd derivative spectra present chemical components in the ground roasted coffee as reported by [43]. The spectra showed peaks of components in coffee that were not only water but also chlorogenic acid, caffeine, trigonelline and carbohydrates. The acquired absorbance peaks show that around 1412–1444 nm is the first overtone of O-H and N-H, which is related to chlorogenic acid and caffeine as shown in Figure 2B [44]. There were peaks of chlorogenic acid at 1100 and 1500 nm which are the second overtone of C-H stretching and the first overtone of C-H stretching [45,46]. The acquired spectra showed a clear peak of caffeine at 1216 nm which is related to molecules’ bonding with the second overtone of C-H stretching [47]. The acquired spectra also showed the peak at around 1366–1378 nm that is correlated to the second overtone of C-H, which is trigonelline, as well as the peak at around 1626–1640 nm concerns to the first overtone of C-H related to trigonelline and caffeine [48]. The peak of carbohydrates showed at around 1584 nm, which has been shown to be associated with the first overtone of the O-H stretch [45].

### 3.3. Characterization of Spectra for Ground Roasted Coffee by FTIRs

Average spectra of pure Arabica and pure Robusta by FTIRs show the peaks of components in ground roasted coffee (Figure 3) where both spectra of pure Arabica and pure Robusta showed peaks at the same wave numbers indicating that the spectra showed the peaks of components in coffee including water, caffeine, lipid, chlorogenic acid and carbohydrates. Water peaks showed at the wavenumber around 3300 cm^−1^ in the region of 3676–3028 cm^−1^ [49,50]. Caffeine peaks showed at the wavenumber around 2852 cm^−1^ and in the region of 1650–1600 cm^−1^ [51,52,53]. Lipid peaks were attributed at the wavenumber around 1744 cm^−1^ and in the region of 2908–2920 cm^−1^ which is related to stretching vibration of the carbonyl (C=O) and stretching asymmetric C-H of CH_2_ groups [53,54]. The peaks of chlorogenic acid, which occurs by esterification between quinic acid and caffeic acid, showed in the range of 1450–1000 cm^−1^ [5]. The peaks of carbohydrates showed in the region of 1500–700 cm^−1^ [5].

### 3.4. PCA

The acquired spectral data of samples from NIR-HSI and FTIRs were used for the principal component analysis in order to determine the potential of classification of pure Arabica and pure Robusta, as well as the Arabica diluted with Robusta. For pure Arabica and pure Robusta, the score plot of PC1 and PC2 by PCA based on spectral data from NIR-HSI (Figure 4A) shows complete separation between groups of pure Arabica and pure Robusta with the variance percentage of 97% and 3% for PC1 and PC2, respectively. While the score plot of PC1 and PC2 by PCA is based on spectral data from FTIRs (Figure 5A). shows overlapping between groups of pure Arabica and pure Robusta with the variation of 95% and 2% for PC1 and PC2, respectively. For pure Arabica and the Arabica with Robusta, the score plot of PC1 and PC2 by PCA based on spectral data from NIR-HSI (Figure 4B) almost separates between groups of Arabica and Arabica with Robusta with the variance percentage of 98% and 1% for PC1 and PC2, respectively. While the score plot of PC1 and PC2 by PCA based on spectral data from (Figure 5B) shows overlapping between groups of Arabica and Arabica with Robusta with the variation of 50% and 11% for PC1 and PC2, respectively. Therefore, the classification results by using PC1 and PC2 showed that spectral data from NIR-HSI had a better potential when compared with FTIRs.

Samples from measurements were divided into the calibration set and the prediction set for qualitative and quantitative analysis (Table 1). Acquired full-spectrum data of NIR-HSI and FTIRs was used in order to establish the models using SVM. For the qualitative study, the classification between samples of pure Arabica (defined as 0) and the Arabica with Robusta (defined as 1) was analyzed. The pure Arabica samples were divided into the same ratio for both sets and the deviation of the concentration of Robusta in both sets was similar. The samples in the calibration set were 206 and the samples in the prediction set were 102 for both NIR-HSI and FTIRs. For the quantitative study, dependent variables were the various concentration levels of Robusta in the samples of the Arabica with Robusta. The deviation of the concentration of Robusta of samples in the calibration set and the prediction set was similar. The samples in the calibration set were 136 and the samples in the prediction set were 66 for both NIR-HSI and FTIRs.

### 3.5. Qualitative Analysis

Due to the effects on spectra during measurement, including noise, scatter and baseline shift, the data obtained from the NIR-HSI and FTIRs were improved by carrying out the pre-treatment methods of smoothing, first derivative, second derivative, MSC, SNV and combined methods in order to reduce those effects and produce the most accurate model.

For qualitative analysis, Figure 6 shows the confusion matrix for classification using SVMC models. Comparison between results of NIR-HSI and FTIRs was investigated resulting in NIR-HSI giving the best result of classification, obtained by preprocessing the spectra before creating the model with SNV, which correctly predict 98.48% (130/132) for class 1 and 100% (74/74) for class 0 in the calibration set and 96.97% (64/66) for class 1 and 100% (36/36) for class 0 in the prediction set. The overall accuracy for prediction was 98.04% (100/102). Using FTIRs, the best result for classification was obtained by preprocessing the spectra before creating the model with the first derivative combined with SNV. Using this method, it was able to correctly predict 95.45% (126/132) for class 1 and 100% (74/74) for class 0 in the calibration set and 95.45% (63/66) for class 1 and 100% (36/36) for class 0 in the prediction set. The overall accuracy for prediction was 97.06% (99/102) (Table 2).

### 3.6. Quantitative Analysis

The same procedures described for qualitative analysis were used for quantitative analysis. The results showed that the spectral data obtained from NIR-HSI and FTIRs were improved by using pre-treatment. For quantitative analysis, the calibration models for predicting the concentration of Robusta were established using SVMR, with NIR-HSI giving the best results for the calibration model that had been obtained by preprocessing spectra data with the first derivative combined with SNV pretreatment method with R_p_^2^ and RMSEP of 0.964 and 5.47%, respectively. When using FTIRs, the best result of the calibration model was obtained by preprocessing spectra data with the first derivative combined with MSC pretreatment method with R_p_^2^ and RMSEP of 0.956 and 6.07%, respectively. The summarized results of the calibration models for predicting the concentration of Robusta in the Arabica with Robusta (Table 3) showed that both techniques could be successfully used for prediction but, the NIR-HSI technique had slightly better accuracy than the FTIRs based on higher R_p_^2^ and lower RMSEP.

Figure 7 shows the scatter plots of samples in the calibration set and prediction set that used SVMR models for predicting the concentration of Robusta compared with the actual concentration by using NIR-HSI and FTIRs. The plots of data were close to the 45° line, which indicated good accuracy of the models for prediction.

## 4. Conclusions

It was shown that for either qualitative and quantitative determinations of the levels of Robusta coffee in samples of Arabica coffee, both near-infrared spectroscopy hyperspectral imaging and Fourier transform infrared spectroscopy had potential. Moreover, this study showed that models using NIR-HSI gave higher predictive accuracy for indicating the differences between pure Arabica and Arabica with Robusta and could also be used to determine the concentration of Robusta in the Arabica with Robusta mixture. It was therefore concluded that the results showed that both techniques (NIR-HSI and FTIRs) have the potential for use in the inspection of roasted ground coffee to determine the respective levels of Arabica and Robusta within a mixture of the two.

## Figures and Tables

**Figure 1 foods-11-03122-f001:**
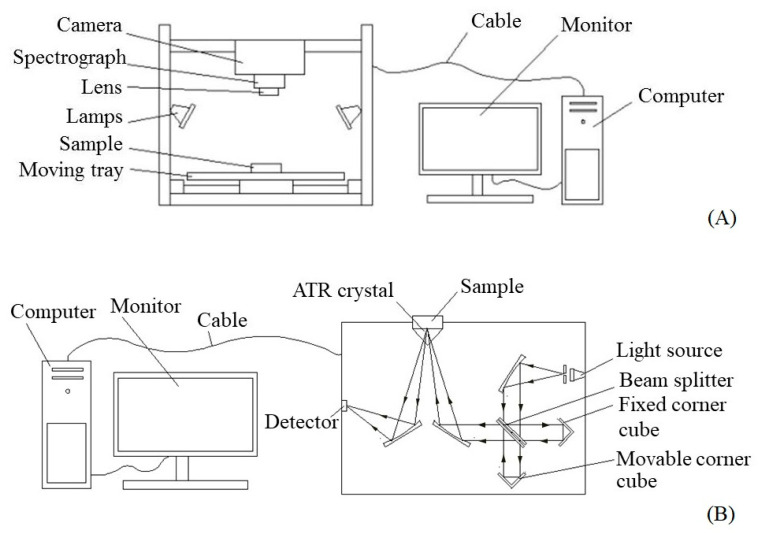
Schematic view of: (**A**) NIR-HSI; and (**B**) FTIRs system.

**Figure 2 foods-11-03122-f002:**
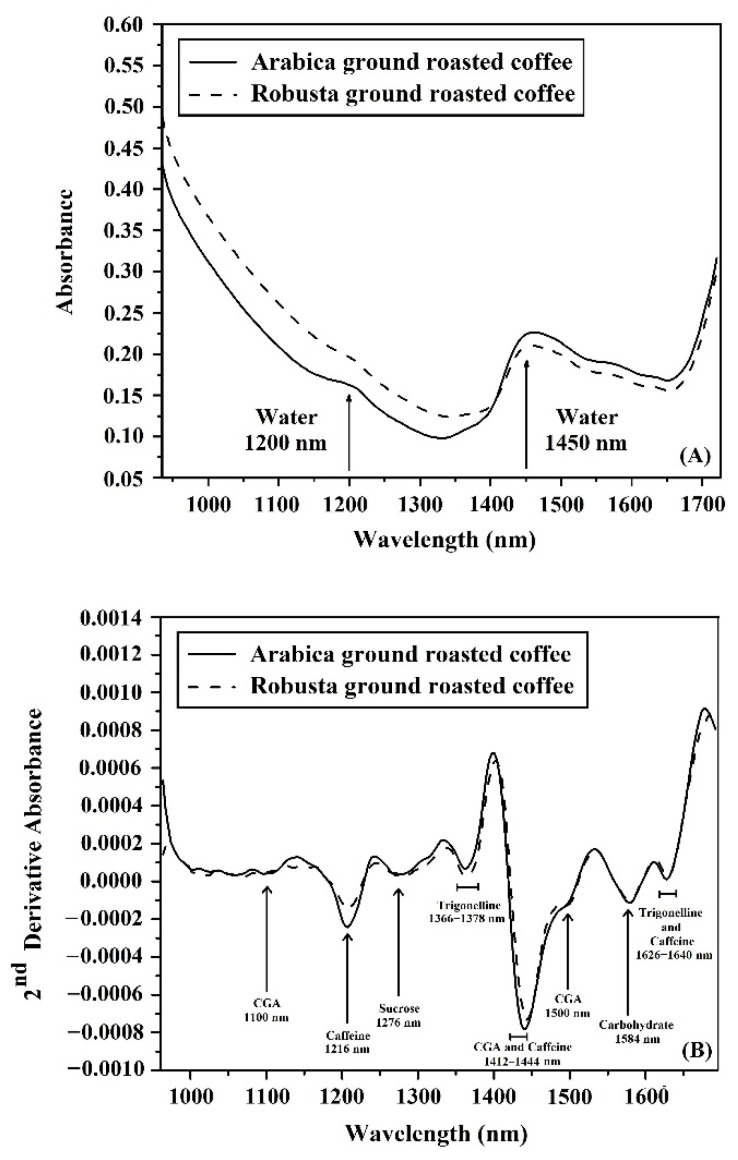
The average absorbance spectra of pure Arabica and pure Robusta using NIR-HSI: (**A**) Original spectra; (**B**) 2nd derivative spectra.

**Figure 3 foods-11-03122-f003:**
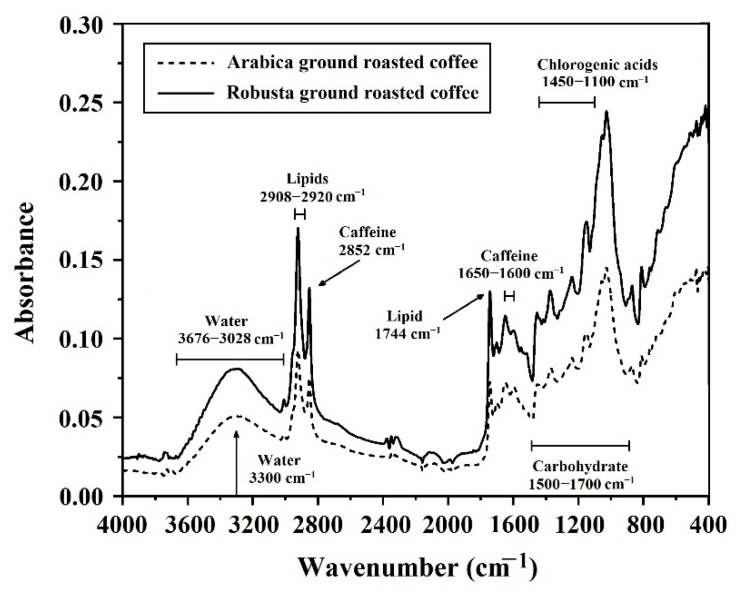
The average original spectra of pure Arabica and pure Robusta using FTIRs.

**Figure 4 foods-11-03122-f004:**
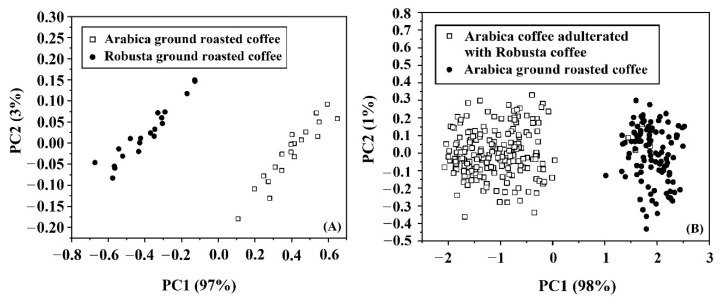
Score plots of PCA for: (**A**) pure Arabica and pure Robusta; (**B**) pure Arabica and Arabica with Robusta using spectral data of NIR-HSI.

**Figure 5 foods-11-03122-f005:**
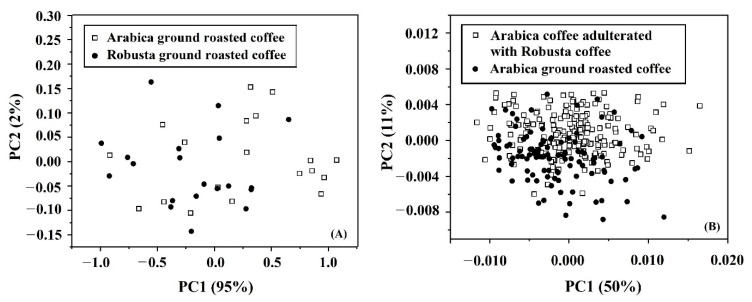
Score plots of PCA for: (**A**) pure Arabica and pure Robusta; (**B**) pure Arabica and Arabica with Robusta using spectral data of FTIRs.

**Figure 6 foods-11-03122-f006:**
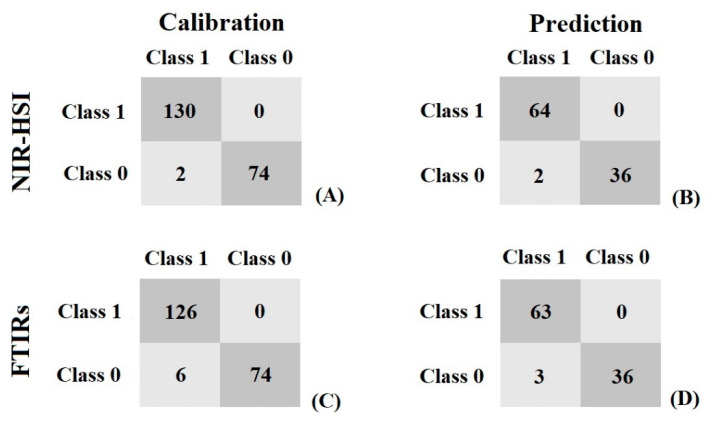
The confusion matrices of classification in the calibration set by NIR-HSI (**A**), the prediction set by NIR-HSI (**B**), the calibration set by FTIRs (**C**), and the prediction set by FTIRs (**D**): Class 0 = pure Arabica, Class 1 = Arabica with Robusta.

**Figure 7 foods-11-03122-f007:**
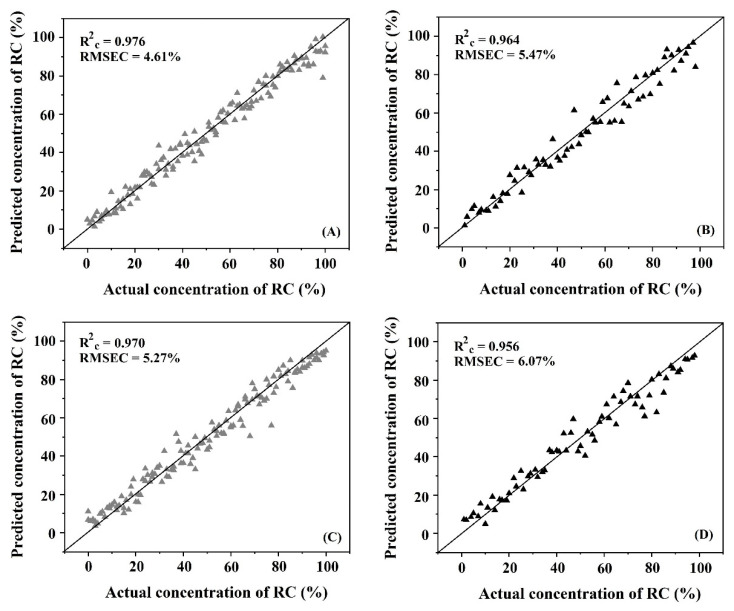
The scatter plot of actual and predicted adulterant concentration in: (**A**) the calibration set by NIR-HSI; (**B**) the prediction set by NIR-HSI; (**C**) the calibration set by FTIRs; (**D**) the prediction set by FTIRs.

**Table 1 foods-11-03122-t001:** The characteristics of samples in the calibration set and the prediction set for SVMC and SVMR using NIR-HSI and FTIRs.

Techniques	Items	Model			
		SVMC ^5^		SVMR ^6^	
		Cal ^7^	Pred ^8^	Cal	Pred
**NIR-HSI ^1^**	Number of samples	206	102	136	66
	% Concentration of Robusta ^3^	0–1	0–1	0–100	1–98
	Mean (%)	0.64	0.65	50.24	49.50
	SD ^4^ (%)	0.48	0.48	29.54	28.79
**FTIRs ^2^**	Number of samples	206	102	136	66
	% Concentration of Robusta	0–1	0–1	0–100	1–98
	Mean (%)	0.64	0.65	50.24	49.50
	SD (%)	0.48	0.48	29.54	28.79

^1^ NIR hyperspectral imaging. ^2^ Fourier transform infrared spectroscopy. ^3^ Support vector machine classification. ^4^ Support vector machine regression. ^5^ Concentration of Robusta ground roasted coffee. ^6^ Standard deviation. ^7^ Calibration. ^8^ Prediction.

**Table 2 foods-11-03122-t002:** Performance of SVMC models for classification between pure Arabica and the Arabica with Robusta using NIR-HSI and FTIRs.

Model	Technique	Number of Samples		Pre-Treatment	% Accuracy		% Specificity		% Sensitivity		% Error Rate	
		Cal	Pred		Cal ^6^	Pred ^7^	Cal	Pred	Cal	Pred	Cal	Pred
**SVMC ^1^**	**NIR-HSI ^2^**	206	102	SNV ^4^	99.03	98.04	100	100	97.37	94.74	0.97	1.96
	**FTIRs ^3^**	206	102	1st derivative ^5^ + SNV	97.09	97.06	100	100	92.50	92.31	2.91	3

^1^ Support vector machine classification. ^2^ NIR hyperspectral imaging. ^3^ Fourier transform infrared spectroscopy. ^4^ Standard normal variate transformation. ^5^ Savitzky-Golay first derivative. ^6^ Calibration. ^7^ Prediction.

**Table 3 foods-11-03122-t003:** Results of SVMR models for detection of concentration of RC in the adulterated AC with RC using NIR-HSI and FTIRs.

Model	Technique	Number of Samples		Pre-Treatment	R^2^_c_	R^2^_cv_	R^2^_p_	RMSEC ^9^ (%)	RMSECV ^10^ (%)	RMSEP ^11^ (%)
		Cal ^4^	Pred ^5^							
**SVMR ^1^**	**NIR-HSI ^2^**	178	88	1st derivative ^6^	0.970	0.953	0.958	4.66	5.86	5.35
	**FTIRs ^3^**	131	63	2nd derivative ^7^ + SNV ^8^	0.965	0.913	0.951	5.85	9.04	6.96

^1^ Support vector machine regression. ^2^ NIR hyperspectral imaging. ^3^ Fourier transform infrared spectroscopy. ^4^ Calibration. ^5^ Prediction. ^6^ Savitzky-Golay first derivative. ^7^ Savitzky-Golay second derivative. ^8^ Standard normal variate transformation. ^9^ Root mean square error of calibration. ^10^ Root mean square error of cross-validation ^11^ Root mean square error of prediction.

## Data Availability

Not applicable.

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
