# Peer review of "Assessing the Levels of Robusta and Arabica in Roasted Ground Coffee Using NIR Hyperspectral Imaging and FTIR Spectroscopy"

_foods, 2022, doi:10.3390/foods11193122_

Round 1

Reviewer 1 Report

This was a well-put together paper and was presented well. However, I do have some concerns about the novelty. Spectroscopy, and specifically infrared spectroscopy, has been EXTENSIVELY used to detect adulteration in foods, and EXTENSIVELY used to detect adulteration in coffee specifically. One paper that lists many of these studies is

Near-Infrared Spectroscopy Applied to the Detection of Multiple Adulterants in Roasted and Ground Arabica Coffee

Foods. 2022 Jan; 11(1): 61. Published online 2021 Dec 28. doi: 10.3390/foods11010061

While this paper may have offered a specific experiment that has not been done before, the novelty/innovation factor remains low.

Specific suggestions: 

Page 1, line 35-36: Some clarification required. The sentence beginning on page 1, line 33 states that Robusta is the main cultivated species (presumably in Africa). However, the following sentence (lines 35-36) state the opposite: that Arabica is the main species cultivated, 60-70%. Perhaps they are talking about a different region or worldwide but this is not clear and as it stands, there is some contradiction here. 

Page 1, line 41: Instead of "substitution of Robusta with Arabica," don't the authors intend "substitution of Arabica with Robusta?"

Reference 16 does seem to contain an introductory paragraph to FTIR, but this is a book chapter on elastomer seals and jet fuels, not anything having to do with FTIR other than a single paragraph. This also has nothing to do with common applications or food application of FTIR. This reference seems to be totally inappropriate, considering that there are many, many introductory texts out there on FTIR and its uses, including ones which explain the science. I question why the authors chose a book chapter about seals and jet fuels with only one paragraph on FTIR. This should be removed and replaced with a more appropriate reference.

Reviewer 2 Report

I consider the manuscript to be relevant and well written. I think the presentation of the Figures in this manuscript can be improved (for instance the schemes of the systems used - FIg.1 and Fig.2 -  could be in one figure separated into A a B parts)
